# Chronic exposure to environmental temperature attenuates the thermal sensitivity of salmonids

Alexia M. González-Ferreras [1,2] ✉, Jose Barquín [1], Penelope S. A. Blyth [3,4], Jack Hawksley[3], Hugh Kinsella[2,5], Rasmus Lauridsen[6,7], Olivia F. Morris[3], Francisco J. Peñas[1], Gareth E. Thomas[2,8], Guy Woodward [3], Lei Zhao[9] & Eoin J. O'Gorman [2]

Metabolism, the biological processing of energy and materials, scales predictably with temperature and body size. Temperature effects on metabolism are normally studied via acute exposures, which overlooks the capacity for organisms to moderate their metabolism following chronic exposure to warming. Here, we conduct respirometry assays in situ and after transplanting salmonid fish among different streams to disentangle the effects of chronic and acute thermal exposure. We find a clear temperature dependence of metabolism for the transplants, but not the in-situ assays, indicating that chronic exposure to warming can attenuate salmonid thermal sensitivity. A bioenergetic model accurately captures the presence of fish in warmer streams when accounting for chronic exposure, whereas it incorrectly predicts their local extinction with warming when incorporating the acute temperature dependence of metabolism. This highlights the need to incorporate the potential for thermal acclimation or adaptation when forecasting the consequences of global warming on ecosystems.

Metabolic rate is commonly described as the transformation of energy or materials within an organism over time[1]. Metabolic rate varies within[2] and among species[3] with the two most important variables thought to be temperature and individual body size[4]. The relationship linking temperature, body size, and metabolic rate has been explained by the Metabolic Theory of Ecology (MTE), with important implications for multiple levels of biological organisation from the individual or cellular level to entire ecosystems[1,5,6].

Temperature is particularly relevant for ectotherms such as fish, whose body temperature is largely regulated by that of their environment[7]. The limited dispersal of freshwater fish within river networks makes them particularly susceptible to global change[8], highlighting the need to understand their responses to warming for better conservation and management. The thermal sensitivity of ectothermic metabolic rate has been widely studied, and it generally rises with temperature over the range an organism normally experiences[9,10]. According to MTE, the slope of the relationship between metabolic rate and temperature (termed the activation energy, $E_A$) is approximately 0.65 eV[1], although it can vary between 0.2 and 1.2 eV[4]. The slope of the relationship between metabolic rate and

[1]IHCantabria - Instituto de Hidráulica Ambiental de la Universidad de Cantabria, C/Isabel Torres 15, 39011 Santander, Spain. [2]School of Life Sciences, University of Essex, Wivenhoe Park, Colchester CO4 3SQ, UK. [3]Georgina Mace Centre for the Living Planet, Department of Life Sciences, Imperial College London, Silwood Park Campus, Buckhurst Road, Ascot SL5 7PY, UK. [4]School of Biosciences, University of Sheffield, Sheffield S10 2TN, UK. [5]Trinity College Dublin, Dublin, Ireland. [6]Game & Wildlife Conservation Trust, Salmon and Trout Research Centre, East Stoke, Wareham BH20 6BB, UK. [7]Six Rivers Iceland, Reykjavik 101, Iceland. [8]Department of Life Sciences, Natural History Museum, Cromwell Road, London SW7 5BD, UK. [9]Beijing Key Laboratory of Biodiversity and Organic Farming, College of Resources and Environmental Sciences, China Agricultural University, Beijing 100193, China. ✉e-mail: gferrerasam@unican.es

body mass (termed the allometric exponent, *b*) has long been assumed to approximate 0.75 across all organisms[1,11,12], however, some studies have questioned the universality of this value, finding large variation among taxa[13,14].

The majority of previous studies focused on metabolic theory and respirometry assays have been conducted in the laboratory, using oxygen consumption rate as a proxy for aerobic metabolic rates[15,16]. Here, organisms are normally transported to the laboratory where they are exposed to acute temperature changes over short time scales (hours to days). However, this might not adequately represent the metabolic rates of individuals experiencing chronic temperature exposure (years to multiple generations), since adaptive responses including thermal acclimation and adaptation might modulate the metabolic process. Here, acclimation occurs through plastic changes such as alteration of phenotypes as a function of the environment with unchanged genotypes, which is often a short-term response within the lifetime of an individual[17], whilst adaptation occurs through evolutionary changes such as alteration of genetic variation, which is often a long-term response across multiple generations[18]. Accordingly, previous studies have revealed plasticity or evolution in metabolic rates of fish[19–21] and aquatic invertebrates[22]. To the best of our knowledge, however, no studies have been conducted in the wild to examine the effects of chronic exposure to warming on the physiological responses of native populations (i.e. experiencing elevated temperature over long time scales, encompassing many generations). Therefore, a lack of field experiments and empirical data means we still have limited knowledge about the extent to which adaptive responses to warming might modulate metabolism in natural environments.

Metabolic theory has been incorporated into models to predict how global change will affect food webs[23,24] or carbon cycling[25,26].

However, these ignore the potential for organisms to adjust their physiological response to warming through phenotypic or evolutionary changes. If adaptive responses to warming can alleviate the energetic demands of organisms (e.g. by downregulating their metabolic rate), then modelling studies that do not account for this are likely to overestimate the impacts of long-term warming. Studies quantifying the effect of chronic exposure to higher temperatures on the thermal sensitivity of metabolic rates are therefore urgently needed to improve our ability to forecast the effects of global warming on ecosystems.

To address this, we measured the metabolism of a widespread cold-water fish species – the brown trout (*Salmo trutta* Linnaeus, 1758) – across multiple streams in the same catchment (Hengill, SW Iceland; Fig. 1). Due to geothermal activity, the streams vary in their mean annual temperature by 3–20 °C, but are otherwise alike in their physicochemical properties, making this a large-scale natural warming experiment. We conducted respirometry assays in situ and after transplanting fish among streams with contrasting temperatures, allowing us to disentangle the effects of chronic and acute thermal exposure (Fig. 1b, c). Additionally, we characterized the metabolism of brown trout and Atlantic salmon (*Salmo salar* Linnaeus, 1758) in situ in three additional locations (UK, Spain, and NE Iceland, spanning temperatures of 6–20 °C) to test the generality of our findings across the full continental-scale latitudinal range of these ubiquitous salmonid species (Fig. 1a). Lastly, we used a bioenergetic population dynamical model[27] to assess the food-web implications of altered thermal sensitivity of fish metabolism for the community biomass of algae, invertebrates, and fish. Our first hypothesis was that metabolic rates increase with both temperature and body mass. Our second hypothesis was that populations experiencing chronic temperature exposure

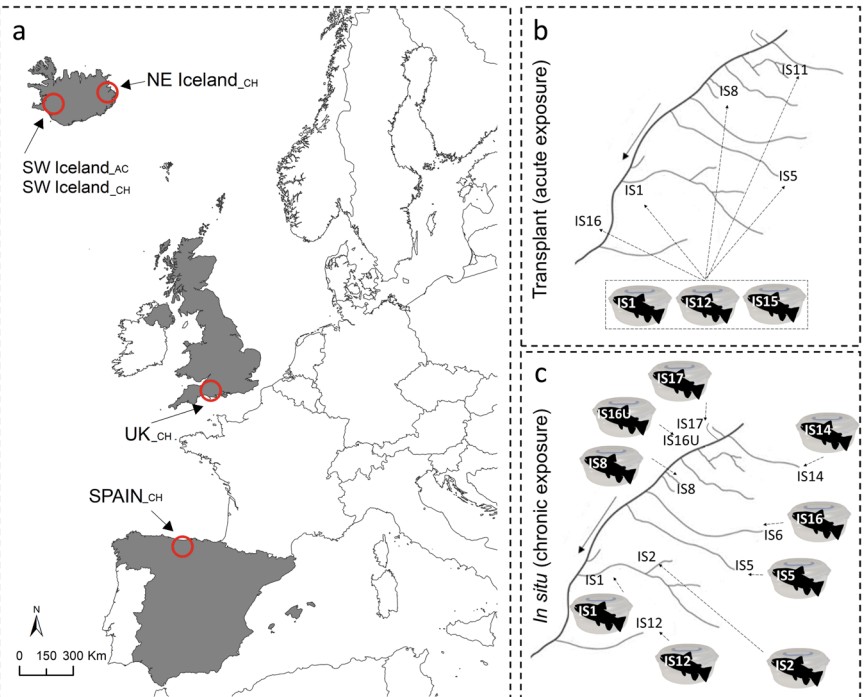

**Fig. 1 | Map of study sites and overview of acute versus chronic exposure assays.** **a** Locations of the study sites, incorporating five assay contexts: SW Iceland_CH, SW Iceland_AC, NE Iceland_CH, UK_CH, and Spain_CH (subscripts indicate whether chronic [CH] or acute [AC] temperature exposures were investigated). **b** Graphical representation of transplant (i.e. acute exposure) and (**c**) in situ (i.e. chronic exposure) fish respiration assays performed in the Hengill geothermal catchment. Streams are labelled with the same code used in previous studies[65] and the solid arrows indicate water flow direction. In both assays, the stream code on the fish icon indicates the river where the fish was caught, and the dashed arrow denotes the river where respirometry assays were conducted. Note that fish from IS1, IS5, and IS12 were not "transplanted" to their own streams. The geographical delimitation for countries in (**a**) has been obtained through Natural Earth (www.naturalearthdata.com) under public domain. The fish silhouette in (**b**) and (**c**) was adapted from an image of *Salmo trutta* (by Carlos Cano-Barbacil) downloaded from PhyloPic (https://www.phylopic.org/) under CC0 1.0 Universal Public Domain Dedication.

**Table 1 | Statistical output of multiple linear regression models for each assay context in the study**

| Species | Assay context | n | Coefficient | Estimate | SE | t value | p value | $r^2$ |
|---|---|---|---|---|---|---|---|---|
| Brown trout | SW Iceland$_{AC}$ | 79 | $\ln(I_0)$ | −0.551 | 0.228 | −2.413 | 0.018 | 0.51 |
| | | | $b$ | 0.628 | 0.079 | 7.942 | <0.001 | |
| | | | $E_A$ | 0.361 | 0.080 | 4.509 | <0.001 | |
| Brown trout | SW Iceland$_{CH}$ | 75 | $\ln(I_0)$ | 0.124 | 0.253 | 0.492 | 0.624 | 0.51 |
| | | | $b$ | 0.591 | 0.067 | 8.842 | <0.001 | |
| | | | $E_A$ | 0.054 | 0.200 | 0.272 | 0.786 | |
| Brown trout | UK$_{CH}$ | 157 | $\ln(I_0)$ | −0.037 | 0.341 | −0.109 | 0.913 | 0.09 |
| | | | $b$ | 0.668 | 0.165 | 4.046 | <0.001 | |
| | | | $E_A$ | −0.443 | 0.477 | −0.929 | 0.354 | |
| Brown trout | Spain$_{CH}$ | 79 | $\ln(I_0)$ | 0.094 | 0.227 | 0.414 | 0.680 | 0.40 |
| | | | $b$ | 0.567 | 0.078 | 7.261 | <0.001 | |
| | | | $E_A$ | 0.150 | 0.357 | 0.421 | 0.675 | |
| Atlantic salmon | Spain$_{CH}$ | 19 | $\ln(I_0)$ | 0.337 | 0.561 | 0.600 | 0.557 | 0.26 |
| | | | $b$ | 0.511 | 0.184 | 2.772 | 0.014 | |
| | | | $E_A$ | −0.015 | 0.895 | −0.017 | 0.987 | |
| Atlantic salmon | NE Iceland$_{CH}$ | 34 | $\ln(I_0)$ | −2.612 | 0.816 | −3.200 | 0.003 | 0.41 |
| | | | $b$ | 1.644 | 0.340 | 4.839 | <0.001 | |
| | | | $E_A$ | 0.062 | 0.432 | 0.143 | 0.887 | |

The estimated coefficients for the intercept ($I_0$), allometric exponent ($b$), and activation energy ($E_A$) are shown with associated standard errors (SE), t values, and p values. Results were obtained from a linear model describing the relationship between routine metabolic rate [ln($I$) in mg O$_2$ h$^{-1}$] as the response variable and fish body mass [ln($M$) in mg] and standardised Arrhenius temperature ($T_A$ in K) as explanatory variables. The $r^2$ value and number of individual fish included for each model ($n$) are also provided.

will be less thermally sensitive than those experiencing acute temperature exposure due to adaptive responses. Our third hypothesis was that any reduction in thermal sensitivity of metabolic rate will support a higher-than-expected biomass of fish in warmer environments.

In this work, we find that chronic exposure to warming can attenuate salmonid thermal sensitivity and we show how a bionergetic model can capture the presence of fish in warmer streams when accounting for chronic exposure.

## Results

Routine metabolic rates (i.e. when individuals exhibit normal activity; see *Methods*) were quantified for a total of 511 individual fish (none of which were ever reused) in field respirometry assays. This included 83 brown trout in SW Iceland$_{AC}$, 90 brown trout in SW Iceland$_{CH}$, 188 brown trout in UK$_{CH}$, 93 brown trout and 22 Atlantic salmon in Spain$_{CH}$, and 35 Atlantic salmon in NE Iceland$_{CH}$, (subscripts indicate whether chronic [CH] or acute [AC] temperature exposures were investigated; see Fig. 1, Table S1, and *Methods* for study site descriptions).

### Temperature-dependence of metabolism

Metabolic rate only increased significantly with temperature in the acute thermal exposure (transplant) assay in SW Iceland, with an activation energy of 0.361 ± 0.160 eV (mean ± 95% CI; Table 1, Fig. 2a). There was no significant effect of temperature on metabolic rate in the chronic thermal exposure (in situ) assay in SW Iceland (0.054 ± 0.399 eV; Table 1, Fig. 2b), nor in any of the other locations (Table 1, Fig. 3a–d). This only agrees with our first hypothesis for the acute exposure assay (i.e. confidence intervals of the activation energy do not include zero), and not the chronic exposure assay (i.e. confidence intervals include zero and thus thermal sensitivity is attenuated). Note that the activation energy of the chronic exposure assay was lower than that of the acute exposure assay, but the confidence intervals overlapped, indicating no clear support for our second hypothesis. Values of activation energies differed from the expected value of 0.65 eV from MTE (including 95% CI) for trout in

SW Iceland$_{CH}$, SW Iceland$_{AC}$, and UK$_{CH}$. Note that supplementary analysis indicated that the source stream from which the fish were collected in the acute thermal exposure assays had no significant effect on metabolic rate (Table S3).

### Size-dependence of metabolism

In support of our first hypothesis, there was a significant log-linear increase in metabolic rate with body mass for all assay contexts (Table 1), regardless of species and whether the thermal exposure was chronic or acute. The allometric exponents obtained for assays on brown trout in the acute and chronic exposures in SW Iceland were 0.628 ± 0.158 (mean ± 95% CI; Fig. 2c) and 0.591 ± 0.133 (Fig. 2d), respectively. For the remaining locations with brown trout presence, scaling exponents ranged from 0.567 ± 0.156 in Spain$_{CH}$ (Fig. 3f) to 0.668 ± 0.326 in UK$_{CH}$ (Fig. 3e). For Atlantic salmon, scaling exponents ranged from 0.511 ± 0.391 in Spain$_{CH}$ (Fig. 3g) to 1.644 ± 0.693 in NE Iceland$_{CH}$ (Fig. 3h). Values of allometric exponents differed from the "universally expected" value of 0.75 from MTE (including 95% CI) for brown trout in SW Iceland$_{CH}$ and Spain$_{CH}$, and for Atlantic salmon in NE Iceland$_{CH}$.

### Food-web implications

The bioenergetic model parameterised with values from our previous work[27] (i.e. optimised for chronic exposure to the natural system and thus assuming $E_{x3}$ = 0.054) explained 32%, 84%, and 97% of the variation across streams in the empirical biomass of diatoms, invertebrates, and fish, respectively (Fig. 4a). Altering only the parameter associated with thermal sensitivity of fish metabolism in the bioenergetic model to the value obtained in the SW Iceland$_{AC}$ (transplant) assays ($E_{x3}$ = 0.361) dramatically changed the predictions, whereby fish were instead predicted to exist in colder streams and go extinct as temperature increased (Fig. 4b). This supports our third hypothesis that a reduction in thermal sensitivity of fish due to chronic exposure would result in a higher than expected biomass of fish in warmer streams. The higher biomass of fish in the colder streams predicted from the acute exposure assays also exerted stronger top-down control on invertebrates, leading to an

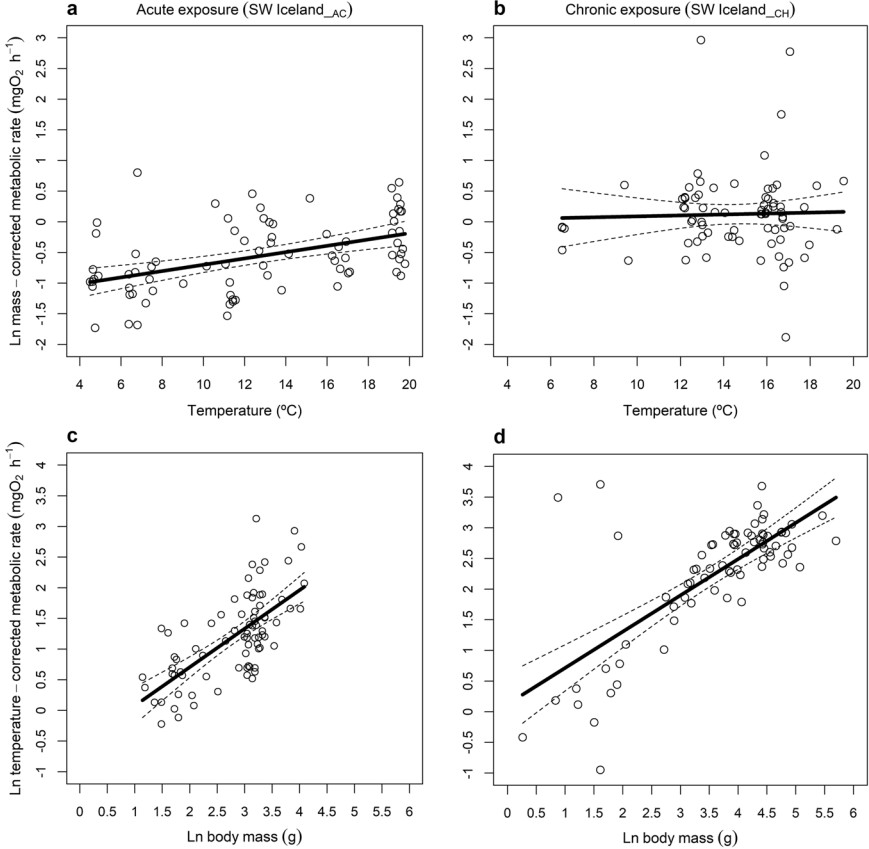

**Fig. 2 | Mass and temperature dependence of salmonid metabolism in the Hengill system.** Effects of (**a**, **b**) temperature and (**c**, **d**) mass on the metabolic rate of brown trout following (**a**, **c**) acute exposure (SW Iceland$_{AC}$) and (**b**, **d**) chronic exposure (SW Iceland$_{CH}$) assays. Linear regressions: (**a**) $\ln(I_M) = -0.551 + 0.361\,T_A$, $F_{1,77} = 20.61$, $p < 0.001$, $r^2 = 0.201$; (**b**) $\ln(I_M) = 0.124 + 0.054\,T_A$, $F_{1,73} = 0.075$, $p = 0.785$, $r^2 = -0.013$; (**c**) $\ln(I_T) = -0.551 + 0.628\ln M$, $F_{1,77} = 63.94$, $p < 0.001$, $r^2 = 0.447$; (**d**) $\ln(I_T) = 0.124 + 0.591\ln M$, $F_{1,73} = 79.27$, $p < 0.001$, $r^2 = 0.514$. Source data are provided as a source data file.

increase in algal biomass, which was reversed in the warmer streams without the fish.

## Discussion

We found a clear temperature dependence of metabolism for the transplant, but not the in-situ assays, suggesting that chronic exposure to warmer environments can attenuate the thermal sensitivity of metabolism in brown trout. This dramatic physiological change suggests that simplistic extrapolations from theory or laboratory studies that do not account for adaptive responses may be of questionable validity when forecasting the consequences of global warming in wild ecosystems.

Studies performed in the laboratory with fish exposed to different experimental temperatures have found activation energies between 0.2 and 1.2 eV[4,28]. We only found mean values of $E_A$ within this range in our transplant assays (SW Iceland$_{AC}$), suggesting that thermal sensitivity may only emerge for brown trout following acute exposure to new temperatures. Remarkably, we did not find any significant relationships between temperature and metabolic rate for any of the assay contexts where we measured fish in situ (i.e. chronic temperature exposure), with values of $E_A \leq 0.15$ eV. In laboratory experiments, fish are usually caught in the wild or from hatcheries and transported to experimental facilities, with the stress of movement from their home environment to artificial conditions one possible explanation for the difference to our results. Additionally, the use of environmental temperatures rather than acute exposures to new thermal regimes suggests that the longer salmonids spend in a warmer environment, the more capable they are of downregulating their metabolism.

The observed attenuation of thermal sensitivity in the chronic exposure assays has major implications for models using a universal temperature dependence of metabolic rate when forecasting the long-term impacts of warming on ecosystems. MTE is based on the laws of thermodynamics[1], and assumes that factors such as thermal acclimation or adaptation can only alter the intercept and not the slope of the relationship between metabolism and temperature[29]. However, some studies have shown that fish populations from warmer environments can display metabolic rates below expectations[30], suggesting that adaptive responses can lower the activation energy[20]. The comparative lowering of respiration rates in warmer environments is referred to as metabolic compensation and temperature-independence of metabolism has been demonstrated for different aquatic organisms such as fish[31] or copepods[32], among others. At the same time, numerous previous studies have shown the temperature-dependence of metabolism in several aquatic organism[10,33] and therefore it is essential to carry out research such as our study to analyse in depth the thermal sensitivity of organism in warmer environments.

Our study differs from previous research in one key aspect: we measured metabolic rates in situ, utilising natural temperature gradients to examine the effects of chronic exposure to a given temperature on metabolic rate. Populations of salmonid fish in the Hengill system are characterised by a high percentage of stationary individuals with very low dispersal beyond their home stream, indicating they will potentially have experienced the same thermal regime over many generations[34]. We are aware of one other study using in-situ respirometry across a wide temperature gradient[20], which found low values of $E_A$ for western mosquitofish, offering some support for adaptive responses reducing the metabolic cost of warming. However, there

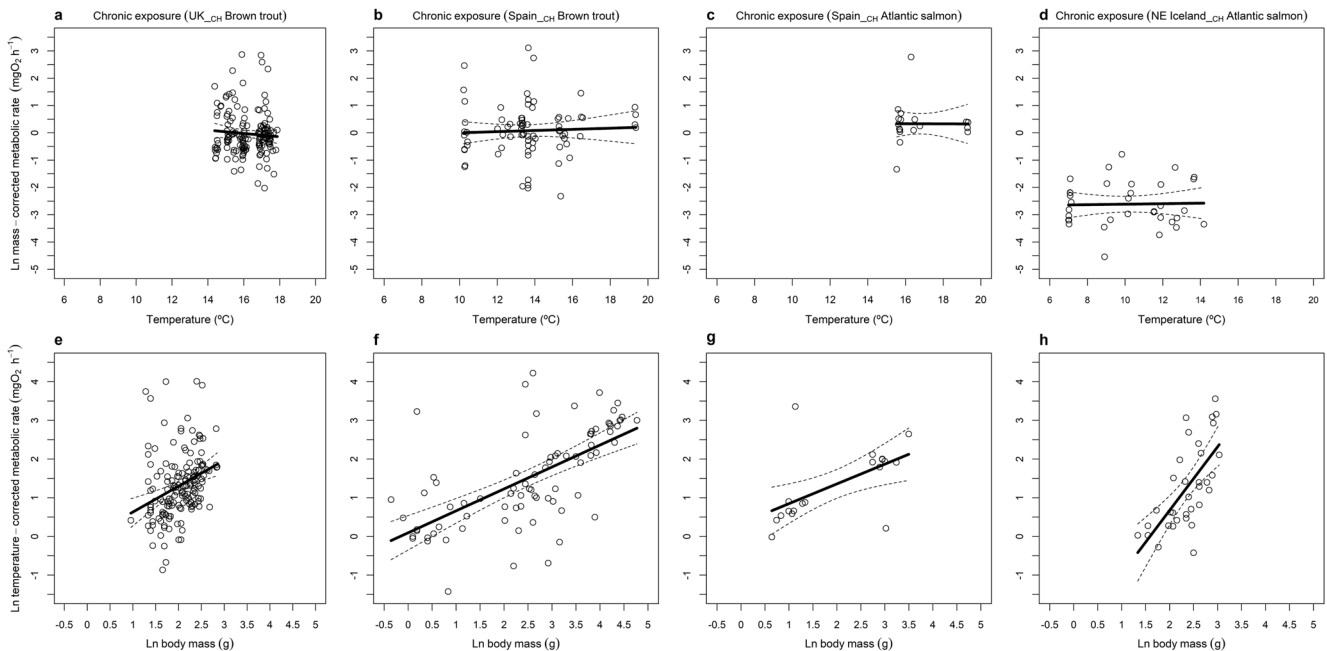

**Fig. 3 | Mass and temperature dependence of salmonid metabolism across the latitudinal gradient.** Effects of (**a**–**d**) temperature and (**e**–**h**) mass on the metabolic rate of brown trout and Atlantic salmon following chronic exposure assays in (**a**, **e**) UK, (**b**, **c**, **f**, **g**) Spain, and (**d**, **h**) NE Iceland. Linear regressions: (**a**) $\ln(I_M) = -0.037 - 0.443\,T_A$, $F_{1,155} = 0.870$, $p = 0.352$, $r^2 = -0.001$; (**b**) $\ln(I_M) = 0.094 + 0.150\,T_A$, $F_{1,77} = 0.180$, $p = 0.673$, $r^2 = -0.011$; (**c**) $\ln(I_M) = 0.337 - 0.015\,T_A$, $F_{1,17} < 0.001$, $p = 0.986$, $r^2 = -0.059$; (**d**) $\ln(I_M) = -2.612 + 0.062\,T_A$, $F_{1,32} = 0.023$, $p = 0.882$, $r^2 = -0.031$; (**e**) $\ln(I_T) = -0.037 + 0.668\ln M$, $F_{1,155} = 16.48$, $p < 0.001$, $r^2 = 0.090$; (**f**) $\ln(I_T) = 0.094 + 0.567\ln M$, $F_{1,77} = 53.42$, $p < 0.001$, $r^2 = 0.402$; (**g**): $\ln(I_T) = 0.337 + 0.511\ln M$, $F_{1,17} = 8.648$, $p = 0.009$, $r^2 = 0.298$; (**h**) $\ln(I_T) = -2.612 + 1.644\ln M$, $F_{1,32} = 25.68$, $p < 0.001$, $r^2 = 0.428$. Source data are provided as a source data file.

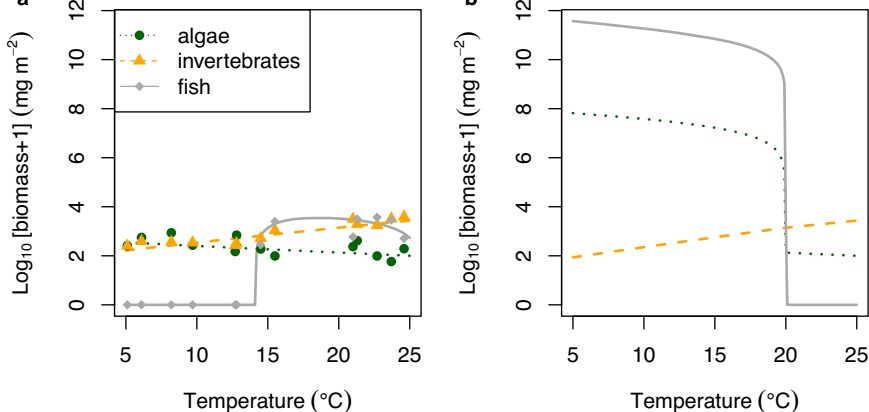

**Fig. 4 | Predicted effects of temperature on biomass of three trophic groups from a bioenergetic model. a** Model predictions (lines) were obtained using the parameter values from our previous work[27], i.e. optimised for chronic exposure to the natural system and thus assuming the same thermal sensitivity of fish metabolic rate in the chronic temperature exposure (in situ) assays in SW Iceland. Empirical biomasses (circles, triangles and squares) of the three major trophic groups in the Hengill streams are also shown, along with mean temperatures during the two-week sampling period[27]. **b** Model predictions obtained using the thermal sensitivity of fish metabolic rate in the acute temperature exposure (transplant) assays in SW Iceland. Source data are provided as a source data file.

was still a significant relationship between temperature and metabolic rate, which may be related to the mosquitofish being an invasive population in the study system, potentially limiting the scope for thermal adaptation over longer time scales[21]. In-situ respirometry in aquatic systems has otherwise only been used to quantify metabolic rates for sharks[35], deep-sea demersal fish[36], corals[37], and salmonids[38], but not across a natural temperature gradient. More in-situ respirometry studies are thus needed and should where possible include population genetics and transcriptomics to determine the relative extent of evolutionary adaptation versus phenotypic plasticity in moderating metabolic responses to warming[39,40].

We observed a positive relationship between metabolic rate and body mass in all our assays with values of $b$ ranging from 0.5 to 1.7. The latter value is much higher than typically reported[41,42], and indicates that larger fish use more oxygen per unit mass than smaller fish, in contrast to the more typical sublinear allometric scaling of metabolic rate. There is still plenty of debate in the literature about whether the theoretical value of $b$ should approximate two-thirds, three-quarters, or linear scaling[13], however, highlighting the inherent variability that can be found. For example, steeper allometric scaling of fish ($b = 0.89$) has been reported from a meta-analysis of 25 studies[43]. Nevertheless, there was a significant positive scaling of metabolic rates with body

mass throughout our results, regardless of species or geographic location, supporting the idea that metabolism consistently exhibits a positive relationship with body mass[13,44].

Our model results accurately captured the long-term response of brown trout to warming across different streams in the Hengill catchment when considering the effects of chronic exposure to warmer environments on metabolism, with a greater biomass of fish as stream temperature increased. This pattern is in contrast to our model predictions based on the acute thermal exposure assays and the general expectation that warming will result in declining body size[45] and loss of apex predators[46], highlighting how a single parameter can completely reverse the expectations of community responses to warming. The surprising success of brown trout in the warmer streams (given it is largely a cold-water species) is most likely driven by increased productivity underpinned by greater nutrient supply as temperature increases[27], helping to meet the greater metabolic demands of a warmer environment. Moreover, trout feed selectively on more energetically valuable prey and benefit from an increased trophic transfer efficiency in the warmer streams[34]. The fact that trout may experience long-term adaptation to stream temperatures over multiple generations in the system highlights the potential for salmonids to adjust their physiology to compensate for warming impacts over the decadal timescales relevant to future global climate change.

There are of course limitations to the approach presented here. Whilst our modelling shows the potential for parameters based on acute versus chronic exposure assays to qualitatively change the effects of temperature on food web dynamics, the quantitative changes in community biomass are rather extreme (i.e. several orders of magnitude in Fig. 4). This may largely be driven by the fact that all other parameters are unlikely to remain equal in a warming scenario. For instance, if global warming alters the temperature dependence of fish metabolic rate (the only parameter we changed here), it should also change the temperature dependence of their feeding rates and indeed the biological rates of lower trophic levels. Previous studies have shown that adaptive responses to warming could influence invertebrate predator–prey interactions and population dynamics[47]. This highlights the importance of quantifying the effects of chronic exposure to warmer environments on the thermal response of biological rates for organisms spanning multiple trophic levels if we are to accurately parameterise predictive models of future warming scenarios on food web dynamics.

Our results emphasise the importance of measuring metabolic rates in situ for a more comprehensive understanding of thermal sensitivity in field conditions[48]. The next key step is to incorporate intermittent flow respirometry in future in-situ studies to quantify basal and maximum metabolic rate (and thus aerobic scope), which would reduce the variability associated with estimates of routine metabolic rate[30,49]. Future studies should evaluate whether there are contrasting adaptive responses depending on the target organism or trophic group, given the different generation times and pace of life for organisms throughout the food web. Our results may be more relevant for high latitude ecosystems where organisms often have scope for increased performance, with potentially different adaptive responses for tropical species close to their thermal limits. Studies are also needed to determine whether adaptation or acclimation can keep pace with, or their extent is sufficient to mitigate, the effects of global climate change. Incorporating thermal acclimation and adaption into predictive models may help to account for some of the ecological surprises in response to warming that have been reported in previous studies[50,51] and help to avoid overestimating the long-term effects of warming on ecosystems.

## Methods

All procedures were performed in accordance with the relevant guidelines and regulations of each country. Icelandic fieldwork was performed in collaboration with the Marine and Freshwater Research Institute under their permits and regulations. Biological field sampling permits were requested in Spain from the regional governments of the study area and the Picos de Europa National Park, and the corresponding authorisation was received. All procedures in the UK were carried out by licenced personnel under a UK Home Office A(SP)A licence (PPL 30/3277).

### Study area

The Hengill geothermal catchment of SW Iceland in May 2018 (transplant assays) and August 2022 (in situ assays; see Fig. 1) was our focal field system (more detailed descriptions can be found in[22,34,52,53]). Headwater streams in the system differ in mean annual temperature from 3–20 °C due to geothermally warmed groundwater, but are otherwise alike in their physical and chemical characteristics[27]. Temperature differences between streams are consistent throughout entire years[27], and over at least a 20-year period of research in Hengill[54], increasing the likelihood that trout (the only fish in the catchment) experience long-term adaptation to streams over multiple generations. Trout populations are composed of a high percentage of stationary (93%; low mobility and dispersal linked to the home range) and low percentage of mobile (7%; high mobility) individuals (based on a mark-recapture study in the Hengill catchment[34]), increasing the possibility for adaptive responses to warming over multiple generations. Thus, this model systems allows us to embed short-term manipulative assays within a long-term temperature gradient[55].

Two types of assays were conducted within the Hengill geothermal streams: (1) **transplant respirometry (SW Iceland_AC)**: fish were collected from cold (IS12 with a mean ± standard deviation annual temperature of $7.8 \pm 4.2\,°C$) and warm streams (IS1 = $11.3 \pm 4.0\,°C$ and IS5 = $13.8 \pm 1.6\,°C$), and their metabolism was measured in five different streams (i.e. transplant of fish between rivers; see Fig. 1b and *Assay procedure* section for more information) to examine the effects of acute thermal exposure; and (2) in situ **respirometry (SW Iceland_CH)**: metabolism was measured in the same nine streams where the fish were caught (i.e. no transplant of fish between streams; see Fig. 1c) to examine the effects of chronic temperature exposure. Due to logistical limitations, the number of streams in the transplant assay was lower than the in-situ assay, but the temperature gradient and number of fish measured was similar across assays (Table S1).

Performing both assays in non-geothermal catchments is complicated by smaller temperature gradients and the logistical difficulty and stress to the fish of implementing transplant assays in rivers that are very far from each other. However, to test the generality of the chronic temperature exposure effects, we conducted in situ assays in three additional locations across the natural latitudinal range of brown trout and its widespread close congener, the Atlantic salmon: (1) UK_CH, consisting of two carriers of the River Frome (East Bourton Boundary Stream and Woodsford North Stream), sampled in August 2021; (2) Spain_CH, consisting of 10 rivers spread evenly across the Deva and Pas catchments, sampled in July 2021; and (3) NE Iceland_CH, consisting of four rivers (Hafralónsá, Hofsá, Selá, and Vesturdalsá) sampled in June 2018 (Fig. 1a).

### Assay procedure

Fish were captured by electrofishing in all study sites and the methodology for field respirometry was consistent. Assay chambers consisted of 7.2 L round (32 cm diameter), airtight, and transparent plastic containers (LocknLock brand), which were submerged in a 50 L container of river water, filtered through a 250 μm sieve (Fig. S2). One individual fish was placed in each chamber and the lid was sealed underwater, ensuring there were no air bubbles in the chamber. Chambers were secured in shallow water for approximately 1.5-3 hours. Each time assays were run, one chamber contained only

filtered river water (i.e. no fish) to act as a control for background photosynthesis and/or respiration of micro-organisms. A miniDOT logger (Precision Measurements Engineering "PME") was inserted into each chamber to measure dissolved oxygen concentrations and water temperature every minute. At the end of each assay, fish were weighed and measured (fork length) and released into the same river from which they were captured. Weights of all fish individuals from Hengill 2018 and 20 individuals from NE_Iceland were estimated according to length-weight relationships obtained from empirical data collected at those locations (see Fig. S1).

## Quantifying metabolic rates

Fish exhibited some activity during the assays since movement is necessary to maintain their position in the water, thus we consider the decline in dissolved oxygen consumption rate as a proxy for routine metabolic rate here[15,56,57]. To reduce potential effects of stress associated with collection and handling of fish, the first 30 minutes of recorded oxygen data in each assay were excluded from further analysis. The maximum duration of the recorded data was also standardised to 120 minutes, resulting in a 90-minute assay measurement period (after excluding the first 30 minutes). Note that the maximum duration of the recorded data was >90 & ≤100 minutes for 1% of assays, >100 & ≤110 minutes for 2% of assays, and >110 & <120 minutes for 8% of assays. The 'auto_rate' function in the 'respR' package v2.0.2[58] of R v4.1.3[59] was used to calculate oxygen depletion rates through a combination of rolling regression and Kernel density estimation (KDE) algorithms[58]. This procedure identifies the most linear portion of the data (representative of routine metabolic rates) through rolling linear regressions of 30-minute time frames. Metabolic rates were only retained for further analysis if the $r^2$ value for the regression was greater than 0.8[60], which was the case for 87% of the data. Background respiration rates were calculated as the slope of the linear regression through the entire 90-minute period of the control chamber and subtracted from the corresponding fish metabolic rates for that assay block. Positive oxygen depletion rates after background correction were excluded from further analysis (which was the case in just one assay). Per volume rates (mg $O_2$ m$^{-1}$ L$^{-1}$) were converted into whole organism rates (mg $O_2$ h$^{-1}$) using the effective volume of the chamber, estimated as total volume minus the volumes of the miniDOT and the fish (assuming a density of 1,000 kg m$^{-3}$)[61].

## Statistical analysis

All statistical analyses were conducted in R v4.1.3[59]. In accordance with MTE[1], metabolic rate, $I$ (mg $O_2$ h$^{-1}$), depends on body mass and temperature as:

$$I = I_0 M^b e^{E_A T_A} \tag{1}$$

where $I_0$ is the intercept, $M$ is the wet weight of fish (g), $b$ is an allometric exponent, $E_A$ is the activation energy (eV), and $T_A$ is a standardised Arrhenius temperature:

$$T_A = \frac{T - T_0}{k T T_0} \tag{2}$$

where $T$ is the temperature of the chamber ($K$), $T_0$ is a normalisation constant set to the average temperature ($K$) of all the chambers for each case study, and $k$ is the Boltzman constant ($8.617 \times 10^{-5}$ eV K$^{-1}$). The average temperature for each chamber was calculated over the same time period selected by the 'auto_rate' function to obtain the metabolic rate. $I_0$ varies depending on the type of organism[5,62], whilst $b$ and $E_A$ are often argued to centre around values of 0.75 and 0.65 eV, respectively[1]. We performed a multiple linear regression

('lm' function in the 'stats' package v4.1.3 of R) on the natural logarithmic transformation of Eq. 1 for each assay context to explore the main effects of temperature and body mass on the metabolic rate. Subsequently, metabolic rates were mass-corrected by dividing by $M^b$ (denoted as $I_M$) and temperature-corrected by dividing by $e^{E_A T_A}$ (denoted as $I_T$) to visualise the independent effects of temperature and body mass on metabolic rate, respectively. Moreover, we performed a supplementary analysis for the acute assays, including an extra categorical variable in Eq. 1 for the main effect of the source stream (S) that the fish were collected from (3 levels: IS1, IS12, and IS5).

## Bioenergetic model

In our previous work[27], we built a bioenergetics model to describe the effects of temperature on the biomass of diatoms, invertebrates, and fish, estimating parameter values by maximum likelihood optimisation based on the observed biomass of the three groups in the Hengill system. Here, we summarise the model and its application to this study, with full details in our previous work[27]. The model is:

$$\frac{dB_1}{dt} = rB_1\left(1 - \frac{B_1}{K}\right) - y_2 B_1 B_2 \tag{3}$$

$$\frac{dB_2}{dt} = e_2 y_2 B_1 B_2 - x_2 B_2 - y_3 B_2 B_3 \tag{4}$$

$$\frac{dB_3}{dt} = e_3 y_3 B_2 B_3 - x_3 B_3 \tag{5}$$

Here, $B_1$, $B_2$, and $B_3$ denote the biomass of diatoms, invertebrates, and fish, respectively (mg m$^{-2}$); $r$ is the maximum mass-specific growth rate of diatoms (day$^{-1}$); $K$ is the carrying capacity (mg m$^{-2}$); $x_i$ is the mass-specific metabolic rate of trophic group $i$ (day$^{-1}$); $y_i$ represents the attack rate of trophic group $i$ (m$^2$ mg$^{-1}$ day$^{-1}$); $e_2 = 0.45$ is the assimilation efficiency when invertebrates consume diatoms[63]; and $e_3 = 0.85$ is the assimilation efficiency when fish consume invertebrates[27].

The equilibrium biomasses in the absence of fish are:

$$\begin{cases} B_1^C = x_2/e_2 y_2 \\ B_2^C = \left(1 - B_1^C/K\right) r/y_2 \\ B_3^C = 0 \end{cases} \tag{6}$$

and the equilibrium biomasses in the presence of fish are:

$$\begin{cases} B_1^D = K\left(1 - B_2^D y_2/r\right) \\ B_2^D = x_3/e_3 y_3 \\ B_3^D = \left(e_2 B_1^D - x_2/y_2\right) y_2/y_3 \end{cases} \tag{7}$$

Whether the fish is present or absent is determined by $\lambda_{3,1\&2} = e_3 y_3 B_2^C - x_3$. If $B_2^C > x_3/e_3 y_3$, i.e. $B_2^C > B_2^D$, then $\lambda_{3,1\&2} > 0$ and fish can invade the equilibrium coexistence state of diatoms and invertebrates. Conversely, if $B_2^C < B_2^D$, then $\lambda_{3,1\&2} < 0$ and fish are absent.

To avoid overfitting, we reduced the number of parameters by letting $G_2 = x_2/y_2$, $G_3 = x_3/y_3$, $H_2 = y_2/r$, and $H_3 = y_2/y_3$. Then the equilibrium biomasses in the absence of fish are:

$$\begin{cases} B_1^C = G_2/e_2 \\ B_2^C = \left(1 - B_1^C/K\right)/H_2 \\ B_3^C = 0 \end{cases} \tag{8}$$

and the equilibrium biomasses in the presence of fish are:

$$\begin{cases} B_1^D = K\left(1 - B_2^D H_2\right) \\ B_2^D = G_3/e_3 \\ B_3^D = \left(e_2 B_1^D - G_2\right)H_3 \end{cases} \qquad (9)$$

Meanwhile, $\lambda_{3,1\&2} = y_3(e_3 B_2^C - G_3)$. Considering $y_3$ is always positive, the presence of fish is determined by $\lambda'_{3,1\&2} = e_3 B_2^C - G_3$.

The temperature dependences of the parameters are:

$$\ln G_2 = A_{G2} + C_{G2} T_A \qquad (10)$$

$$\ln G_3 = A_{G3} + C_{G3} T_A \qquad (11)$$

$$\ln H_2 = A_{H2} + C_{H2} T_A \qquad (12)$$

$$\ln H_3 = A_{H3} + C_{H3} T_A \qquad (13)$$

$$\ln K = A_K + C_K T_A \qquad (14)$$

The values of $A_{G2}$, $A_{G3}$, $A_{H2}$, $A_{H3}$, $A_K$, $C_{G2}$, $C_{G3}$, $C_{H2}$, $C_{H3}$, and $C_K$ were estimated in our previous work[27] using the maximum likelihood optimisation. The estimated values are listed in Table S2. Substituting these estimates into Eqs. 8 and 9, we can simulate the equilibrium biomass of the three groups.

In the Hengill system, we can consider fish to be chronically exposed to the temperature of each stream, and thus the activation energy of metabolic rate should be identical to the chronic temperature assays conducted here, i.e. 0.054 (Table 1). Note that the bioenergetic model does not directly use the parameter value of $E_{x3}$, but rather a combination of $E_{x3}$ and $E_{y3}$ (since $G_3 = x_3/y_3$ and $C_{G3} = E_{x3} - E_{y3}$). Thus, we cannot isolate $E_{x3}$ to verify its exact value from the optimisation against the empirical biomasses in our previous work[27], and so must assume that it is 0.054. We can then examine how a change in the activation energy of fish metabolism from the value in the chronic to the acute temperature exposure assays (i.e. from 0.054 to 0.361) alters the biomass of fish and the lower trophic level groups. That is, we increase $C_{G3}$ by $0.361 - 0.054 = 0.307$. The remaining parameters in the model were maintained at the same values as in our previous work[27].

### Reporting summary

Further information on research design is available in the Nature Portfolio Reporting Summary linked to this article.

## Data availability

The data generated in this study[64] have been deposited with the University of Essex Research Data Repository at http://researchdata.essex.ac.uk/189/. Source data are provided with this paper.

## Code availability

Code used to assess the data and generate the figures and tables[64] have been deposited with the University of Essex Research Data Repository at http://researchdata.essex.ac.uk/189/.

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

## Acknowledgements

This study was part of the R&D project WATERLANDS, code PID2019-107085RB-I00, funded by MCIN/AEI/10.13039/501100011033 (J.B "PI"), the R&D project RIFFLE PID2020-114427RJ-I00 funded by MCIN/AEI/10.13039/501100011033 (F.J.P. "PI", A.M.G.F. "Team researcher"), and NERC (NE/L011840/1, E.J.O.G. "PI" and NE/M020843/1, G.W. "PI", E.J.O.G. "Co-Investigator"). Alexia María González-Ferreras acknowledges the financial support from the Government of Cantabria through the Fénix Programme. G.W., E.J.O.G., O.F.M. and R.L. acknowledge the financial support received from Six Rivers Iceland. We thank everyone involved in the field data collection.

## Author contributions

A.M.G.F., E.J.O.G., G.W., J.B., and F.J.P. were responsible for funding. E.J.O.G., A.M.G.F., G.W., and J.B. designed the research work. A.M.G.F., J.B., P.S.A.B., J.H., H.K., R.L., O.F.M., F.J.P., G.E.T., G.W., L.Z. and E.J.O.G. participated in field sampling design and field data collection. A.M.G.F., E.J.O.G and L.Z. analysed the data. A.M.G.F. wrote the first draft with a key contribution from E.J.O.G.. A.M.G.F., J.B., P.S.A.B., J.H., H.K., R.L., O.F.M., F.J.P., G.E.T., G.W., L.Z. and E.J.O.G. revised and contributed to the final draft of the manuscript.

## Competing interests

The authors declare no competing interest.
