## [Peer Review File · Nature Communications]

Chronic exposure to environmental temperature attenuates the thermal sensitivity of salmonidsREVIEWER COMMENTS

Reviewer #1 (Remarks to the Author):

González-Ferreras and coauthors present research on the differences between chronic and acute temperature exposure on the thermal sensitivity of metabolism in two salmonid species, brown trout and atlantic salmon. They perform a cross-site comparison of the temperature sensitivity for acclimated/adapted populations as a confirmation of the observed insensitivity to temperature in chronically exposed organisms. Lastly, they re-parameterize a multi-trophic bioenergetic model to contrast estimated equilibrium biomass patterns when trout temperature sensitivity is parameterized from acute vs chronic exposure estimates. This work provides an important contribution to the potential ecosystem- and landscape-level effects of ectotherm temperature adaptation.

The apparent temperature insensitivity of metabolism among populations exposed to 'chronic' temperature regimes is very interesting and showcases the importance of incorporating acclimation/adaptation into the predicted longer-term effects of climate warming on ecosystem patterns, species persistence, species distributions, etc. This work adds to growing body of evidence bringing nuance to the assumptions found within influential conceptual frameworks such as the Metabolic Theory of Ecology. The in situ respiration data cross-regional and cross-species data presented here provide import field measurement that counters many patterns observed in short-term respiration experiments, but supports other recent work on temperature adaptation (e.g. Moffett et al. 2018).

The implication that the temperature insensitivity of brown trout contributes to their presence/absence and biomass in the Iceland geothermal watershed (contrasted with e.g., only temperature-resource interactions as suggested in O'Gorman et al. 2017) is a potentially important and noteworthy result. The qualitative differences among the bioenergetic models with chronic versus acute temperature sensitivities are quite drastic. In the context of other work from this group, this study builds a developing picture where energy supply (invert biomass) controls the lower temperature bounds on trout distribution and acclimation sets, or extends, the upper temperature bounds. The generalities to ecosystems beyond the Hengill watershed that are more diverse trophically and

taxonomically is potentially impactful and these results may inspire such additional work. However, I do see apparent discrepancies between the data used here and those in O'Gorman et al. 2017 that need further elaboration in the current manuscript.

It seems that the model shown in Figure 4a of the current manuscript is identical to that of O'Gorman et al. 2017? There is qualitative similarities (taking into account the natural log vs log₁₀ scales) and they are quantitatively identical in the variance accounted for in the empirical data; both account for 32%, 84%, and 97% of the biomass across trophic levels. Without doing a complete, formal interrogation of the model it is difficult to understand exactly what might be at play here, but I find it intriguing that the models show such quantitative similarities with apparently much different values of the temperature dependence of trout metabolic rates. Is this the case? Does this arise because the model is generally insensitive to the assumed temperature dependence? Figure 4b suggests otherwise, unless there is a qualitative threshold of trout metabolism temperature sensitivity. Or does Fig. 4b arise because the temperature dependence of metabolic rate was altered while the temperature dependence of other rates (e.g. attack rate) was kept fixed? I had trouble navigating this when tracking the combined parameters, e.g., G_3 , C_{G3} , through the model. If this is the case, I feel a discussion and justification for using different temperature dependencies of biological processes is appropriate here (e.g. citing Dell et al. 2011 here rather than in the introduction). or am I misunderstanding?

Regardless, I think the discussion on pgs. 9-10 in the penultimate paragraph which highlights the previous work in O'Gorman et al. 2017 and O'Gorman et al. 2016 is lacking depth. It pulls heavily from the bioenergetic model in O'Gorman et al. 2017 (and may be identical) and brings new and interesting data, but they both attempt to explain a similar pattern, that of higher abundance of trout in warmer streams (not to be superficial with the interesting work done here). And this I think is the key to the novelty in this work, the connection of in situ metabolic measurements to a bioenergetic ecosystem model is potentially very powerful, however, without a more complete and clear accounting for the very strong similarities between the two models---that of O'Gorman et al 2017 and this manuscript---the very stark differences shown in Figure 4a & b stand upon other, largely unstated assumptions about different temperature sensitivities of biological rates. A more in depth

contrast of O'Gorman 2017 and this work would provide necessary context for this work. Lacking that, I question the robustness of the conclusions without a clearer elaboration on how parameterizing from field data actually effect the model outcomes in the initial models.

Minor Notes:

Is the citation to Dell et al. (28) in the final introductory paragraph meant to reference O'Gorman et al. 2017 (27)? Otherwise the citation sequence is out of order, I believe.

Reviewer #2 (Remarks to the Author):

This study tested for effects of acute versus multigenerational exposure to temperature on the temperature-dependence of routine metabolic rate (hereafter "TDM") in two salmonid species. The authors compared TDM for acute exposure (transplant) versus chronic exposure (in situ) in brown trout in one river system and found the former was significantly different from zero while the latter was not. The authors further tested in situ TDM in four other river systems total (two with atlantic salmon, two with brown trout) and found that TDM for each of the four was nonsignificantly different from zero. The authors concluded that multigenerational exposure negates TDM in salmonids.

Comments

L118, Fig 2: TDM from acute exposure was significantly different from zero, while TDM for chronic exposure was not. However, the CIs for both exposure types overlap substantially (acute lower = $0.361 - 0.160 = 0.201$; chronic upper = $0.054 + 0.399 = 0.453$), so the main conclusion that multigenerational exposure negates TDM does not appear to be supported.

Throughout: The chronic exposure treatment tested for TDM of fish in situ. It was not a manipulation, and thus not an experiment. Consider using "chronic exposure assay" or other language instead of "chronic exposure experiment" to clarify the methods.

L333: Please clarify the source of the fish used in the acute exposure experiment. If two

sources (warm and cold) were used, why not test for an effect of source temperature on TDM?

Minor Comments

Figures: Please set fixed axis ranges within each figure for comparison among the multiple panels/facets.

Figure 2: Or, instead of the above, consider plotting data and regressions from both river systems in the same panel so lines can be readily compared (i.e. combine a with b AND c with d).

L82: Phenotypic change is not necessarily short term and evolutionary change is not necessarily long term.

L334: Is the variation (+/-) here the range, standard deviation, or ?

L369: Please clarify the source of data for length-weight relationships.

REVIEWER COMMENTS

Reviewer #1 (Remarks to the Author):

González-Ferreras and coauthors present research on the differences between chronic and acute temperature exposure on the thermal sensitivity of metabolism in two salmonid species, brown trout and atlantic salmon. They perform a cross-site comparison of the temperature sensitivity for acclimated/adapted populations as a confirmation of the observed insensitivity to temperature in chronically exposed organisms. Lastly, they re-parameterize a multi-trophic bioenergetic model to contrast estimated equilibrium biomass patterns when trout temperature sensitivity is parameterized from acute vs chronic exposure estimates. This work provides an important contribution to the potential ecosystem- and landscape-level effects of ectotherm temperature adaptation.

The apparent temperature insensitivity of metabolism among populations exposed to 'chronic' temperature regimes is very interesting and showcases the importance of incorporating acclimation/adaptation into the predicted longer-term effects of climate warming on ecosystem patterns, species persistence, species distributions, etc. This work adds to growing body of evidence bringing nuance to the assumptions found within influential conceptual frameworks such as the Metabolic Theory of Ecology. The *in situ* respiration data cross-regional and cross-species data presented here provide import field measurement that counters many patterns observed in short-term respiration experiments, but supports other recent work on temperature adaptation (e.g. Moffett et al. 2018).

The implication that the temperature insensitivity of brown trout contributes to their presence/absence and biomass in the Iceland geothermal watershed (contrasted with e.g., only temperature-resource interactions as suggested in O'Gorman et al. 2017) is a potentially important and noteworthy result. The qualitative differences among the bioenergetic models with chronic versus acute temperature sensitivities are quite drastic. In the context of other work from this group, this study builds a developing picture where energy supply (invert biomass) controls the lower temperature bounds on trout distribution and acclimation sets, or extends, the upper temperature bounds. The generalities to ecosystems beyond the Hengill watershed that are more diverse trophically and taxonomically is potentially impactful and these results may inspire such additional work. However, I do see apparent discrepancies between the data used here and those in O'Gorman et al. 2017 that need further elaboration in the current manuscript.

Response #1: Thank you for the kind praise of our study and for your constructive suggestions to improve the manuscript, which we have addressed below.

It seems that the model shown in Figure 4a of the current manuscript is identical to that of O'Gorman et al. 2017? There is qualitative similarities (taking into account the natural log vs log10 scales) and they are quantitatively identical in the variance accounted for in the empirical data; both account for 32%, 84%, and 97% of the biomass across trophic levels. Without doing a complete, formal interrogation of the model it is difficult to understand exactly what might be at play here, but I find it intriguing that the models show such quantitative similarities with apparently much different values of the temperature dependence of trout metabolic rates. Is this they case? Does this arise because the model is generally insensitive to the assumed temperature dependence? Figure 4b suggests otherwise, unless there is a qualitative threshold of trout metabolism temperature sensitivity. Or does Fig. 4b arise because the temperature dependence of metabolic rate

was altered while the temperature dependence of other rates (e.g. attack rate) was kept fixed? I had trouble navigating this when tracking the combined parameters, e.g., G_3 , C_{G3} , through the model. If this is the case, I feel a discussion and justification for using different temperature dependencies of biological processes is appropriate here (e.g. citing Dell et al. 2011 here rather than in the introduction). or am I misunderstanding?

Response #2: Yes, the baseline model in Figure 4a is the same as the one presented in O’Gorman et al. (2017), which we already know does a very good job of predicting the community biomass of the major trophic groups across the temperature gradient in our study system (as evidenced by the high percentage of variation in biomass explained for each group). We then use the model to demonstrate how dramatically those predictions would change if we were to instead parameterize it with the temperature dependence of fish metabolism (E_{x3}) from our acute assays. Please note that we cannot isolate the parameter E_{x3} in the bioenergetic model because it is combined with E_{y3} , i.e. $G_3 = x_3/y_3$ and $C_{G3} = E_{x3} - E_{y3}$. Therefore, we cannot verify that the original model takes the exact same value of E_{x3} as our chronic exposure assays and must assume this, given that all the parameters are optimized to fit the real system (where organisms experience chronic exposure to environmental temperatures, just as they do in our chronic exposure assays). We now make this much clearer in the revised methods (Ln458), results (Ln149), and Figure 4 legend (Ln303).

Ln458: *“In the Hengill system, we can consider fish to be chronically exposed to the temperature of each stream, and thus the activation energy of metabolic rate should be identical to the chronic temperature assays conducted here, i.e. 0.054 (Table 1). Note that the bioenergetic model does not directly use the parameter value of E_{x3} , but rather a combination of E_{x3} and E_{y3} (since $G_3 = x_3/y_3$ and $C_{G3} = E_{x3} - E_{y3}$). Thus, we cannot isolate E_{x3} to verify its exact value from the optimization against the empirical biomasses in our previous work²⁷, and so must assume that it is 0.054. We can then examine how a change in the activation energy of fish metabolism from the value in the chronic to the acute temperature exposure assays (i.e. from 0.054 to 0.361) alters the biomass of fish and the lower trophic level groups. That is, we increase C_{G3} by $0.361 - 0.054 = 0.307$. The remaining parameters in the model were maintained at the same values as in our previous work²⁷.”*

Ln149: *“The bioenergetic model parameterised with values from our previous work²⁷ (i.e. optimized for chronic exposure to the natural system and thus assuming $E_{x3} = 0.054$) explained 32%, 84%, and 97% of the variation across streams in the empirical biomass of diatoms, invertebrates, and fish, respectively (Fig. 4a).”*

Ln303: *“Model predictions (lines) obtained using the parameter values from our previous work²⁷, i.e. optimized for chronic exposure to the natural system and thus assuming the same thermal sensitivity of fish metabolic rate in the chronic temperature exposure (in situ) assays in SW Iceland.”*

Regardless, I think the discussion on pgs. 9-10 in the penultimate paragraph which highlights the previous work in O’Gorman et al. 2017 and O’Gorman et al. 2016 is lacking depth. It pulls heavily from the bioenergetic model in O’Gorman et al. 2017 (and may be identical) and brings new and interesting data, but they both attempt to explain a similar pattern, that of higher abundance of trout in warmer streams (not to be superficial with the interesting work done here). And this I think is the key to the novelty in this work, the

connection of in situ metabolic measurements to a bioenergetic ecosystem model is potentially very powerful, however, without a more complete and clear accounting for the very strong similarities between the two models---that of O'Gorman et al 2017 and this manuscript---the very stark differences shown in Figure 4a & b stand upon other, largely unstated assumptions about different temperature sensitivities of biological rates. A more in depth contrast of O'Gorman 2017 and this work would provide necessary context for this work. Lacking that, I question the robustness of the conclusions without a clearer elaboration on how parameterizing from field data actually effect the model outcomes in the initial models.

Response #3: Hopefully our previous response has clarified any confusion about the bioenergetic model, whereby we visualise food web dynamics using the exact same parameters from O'Gorman et al. (2017) in Figure 4a, and the change in dynamics after tweaking only the parameter associated with temperature dependence of fish metabolic rate in Figure 4b. We now reinforce this in the highlighted paragraph of discussion, noting the dramatic change in predicted community responses to warming based on this single parameter that may be inappropriately estimated through acute thermal exposures in many experimental studies (Ln223). We have also expanded the discussion here as recommended, highlighting the potential power of combining bioenergetic models with *in situ* assays, whilst acknowledging some of the limitations of the current approach and the need for assessing how chronic exposure to warming could alter the biological rates of other trophic groups in the food web (Ln236).

Ln223: *“This pattern is in contrast to our model predictions based on the acute thermal exposure assays and the general expectation that warming will result in declining body size⁴² and loss of apex predators⁴³, highlighting how a single parameter can completely reverse the expectations of community responses to warming.”*

Ln236: *“There are of course limitations to the approach presented here. Whilst our modelling shows the potential for parameters based on acute versus chronic exposure assays to qualitatively change the effects of temperature on food web dynamics, the quantitative changes in community biomass are rather extreme (i.e. several orders of magnitude in Fig. 4). This may largely be driven by the fact that all other parameters are unlikely to remain equal in a warming scenario. For instance, if global warming alters the temperature dependence of fish metabolic rate (the only parameter we changed here), it should also change the temperature dependence of their feeding rates and indeed the biological rates of lower trophic levels. Previous studies have shown that adaptive responses to warming could influence invertebrate predator–prey interactions and population dynamics⁴⁴. This highlights the importance of quantifying the effects of chronic exposure to warmer environments on the thermal response of biological rates for organisms spanning multiple trophic levels if we are to accurately parameterise predictive models of future warming scenarios on food web dynamics.”*

Minor Notes:

Is the citation to Dell et al. (28) in the final introductory paragraph meant to reference O'Gorman et al. 2017 (27)? Otherwise the citation sequence is out of order, I believe.

Response #4: Thank you for detecting this citation error in the final introductory paragraph, which we have now corrected. We have checked all the other references and cannot see any further issues.

Reviewer #2 (Remarks to the Author):

This study tested for effects of acute versus multigenerational exposure to temperature on the temperature-dependence of routine metabolic rate (hereafter "TDM") in two salmonid species. The authors compared TDM for acute exposure (transplant) versus chronic exposure (in situ) in brown trout in one river system and found the former was significantly different from zero while the latter was not. The authors further tested in situ TDM in four other river systems total (two with atlantic salmon, two with brown trout) and found that TDM for each of the four was nonsignificantly different from zero. The authors concluded that multigenerational exposure negates TDM in salmonids.

Response #5: Thank you for the assessment of our paper and the helpful comments which we respond to below.

Comments

L118, Fig 2: TDM from acute exposure was significantly different from zero, while TDM for chronic exposure was not. However, the CIs for both exposure types overlap substantially (acute lower = $0.361 - 0.160 = 0.201$; chronic upper = $0.054 + 0.399 = 0.453$), so the main conclusion that multigenerational exposure negates TDM does not appear to be supported.

Response #6: We broadly disagree with this comment, but appreciate that we need to be much clearer about the logic for it in the manuscript. The main conclusion that chronic exposure to warmer environments negates thermal sensitivity does not make any comparison between acute and chronic exposures – it simply refers to the chronic exposure assays. In other words, this statement relates to our first hypothesis, which is that metabolic rate increases with temperature. It does not do that for the chronic exposure assays because the confidence intervals include zero, and thus thermal sensitivity of metabolic rate is negated in that circumstance. Our second hypothesis is concerned with the comparison between chronic and acute exposures, and you are correct that the confidence intervals overlap, and so we cannot state that chronic exposure reduces the thermal sensitivity of metabolic rate relative to the acute exposure assays. We have been far more explicit about this point in the results.

Ln125: *“This only agrees with our first hypothesis for the acute exposure assay (i.e. confidence intervals of the activation energy do not include zero), and not the chronic exposure assay (i.e. confidence intervals include zero and thus thermal sensitivity is negated). Note that the activation energy of the chronic exposure assay was lower than that of the acute exposure assay, but the confidence intervals overlapped, indicating no clear support for our second hypothesis.”*

Throughout: The chronic exposure treatment tested for TDM of fish in situ. It was not a manipulation, and thus not an experiment. Consider using "chronic exposure assay" or other language instead of "chronic exposure experiment" to clarify the methods.

Response #7: Thank you for this insightful comment. Based on the differences between experimental and observational approaches (Ludwig & Reynolds, 1988), we agree that chronic exposure is not an experiment *per se*. It is an observational approach since the measurements were taken over a range of conditions imposed by nature rather than by the researchers. The term “assay” is more open and can apply to both experimental and

observational studies, thus we have changed the terminology for both chronic and acute exposures from “experiment” to “assay” throughout the text.

Ludwig, J. A., & Reynolds, J. F. (1988). *Statistical ecology: a primer in methods and computing* (Vol. 1). John Wiley & Sons.

L333: Please clarify the source of the fish used in the acute exposure experiment. If two sources (warm and cold) were used, why not test for an effect of source temperature on TDM?

Response #8: This is a valuable suggestion. The source of the fish used in the acute exposure assays was already identified in Figure 1 and the “*Study area*” section of the methods text. However, we conducted an additional analysis to explicitly test whether source temperature had any effect on metabolic rate and found no significant effect (Ln413). We have added the statistical output from this analysis as Table S3 in the Supporting Information and we report the take-home message in the results (Ln133).

Ln413: “*Moreover, we performed a supplementary analysis for the acute assays, including an extra categorical variable in Equation 1 for the main effect of the source stream (S) that the fish were collected from (3 levels: IS1, IS12, and IS5)*”.

Ln133: “*Note that supplementary analysis indicated that the source stream from which the fish were collected in the acute thermal exposure assays had no significant effect on metabolic rate (Table S3).*”

Table S3: “*Statistical output of multiple linear regression model describing the relationship between routine metabolic rate [$\ln(I)$ in $\text{mg O}_2 \text{ h}^{-1}$] as the response variable and fish body mass [$\ln(M)$ in mg], standardised Arrhenius temperature (T_A in K), and the source stream that the fish were collected from (S ; with 3 levels: IS1, IS12 and IS5) as explanatory variables*”.

Effects	DF	Sum of squares	Mean square	F value	p value
$\ln(M)$	1	17.060	17.060	61.999	<0.001
T_A	1	5.631	5.631	20.463	<0.001
S	2	0.689	0.344	1.252	0.292
Residuals	74	20.362	0.275		

Minor Comments

Figures: Please set fixed axis ranges within each figure for comparison among the multiple panels/facets.

Response #9: We now use fixed axis ranges for Figures 2 and 3. We had already done this for Figure 4 and it is not applicable to Figure 1.

Figure 2: Or, instead of the above, consider plotting data and regressions from both river systems in the same panel so lines can be readily compared (i.e. combine a with b AND c with d).

Response #10: The various assay contexts are independent of one another because they were conducted at separate times and/or in separate locations. Thus, it does not make

sense to combine them in a single figure because they also cannot be analysed with a single statistical model. We have gone with the suggestion in the previous comment for fixed axis ranges.

L82: Phenotypic change is not necessarily short term and evolutionary change is not necessarily long term.

Response #11: We have removed the short- and long-term phrasing (Ln84). Moreover, we have modified the explanation of phenotypic and evolutionary changes in the previous paragraph (Ln71).

Ln84: *“However, these ignore the potential for organisms to adjust their physiological response to warming through phenotypic or evolutionary changes.”*

Ln71: *“Here, acclimation occurs through plastic changes such as alteration of phenotypes as a function of the environment with unchanged genotypes, which is often a short-term response within the lifetime of an individual¹⁷, whilst adaptation occurs through evolutionary changes such as alteration of genetic variation, which is often a long-term response across multiple generations¹⁸.”*

L334: Is the variation (+/-) here the range, standard deviation, or ?

Response #12: The variation referred to here is the standard deviation. We have included this information in the revised manuscript:

Ln330: *“...fish were collected from cold (IS12 with a mean \pm standard deviation annual temperature of 7.8 ± 4.2 °C) and warm streams (IS1 = 11.3 ± 4.0 °C and IS5 = 13.8 ± 1.6 °C)...”*

L369: Please clarify the source of data for length-weight relationships.

Response #13: We have clarified the source of the data in both the main text (Ln362) and the legend of Figure S1.

Ln362: *“Weights of all fish individuals from Hengill 2018 and 20 individuals from NE_Iceland were estimated according to length-weight relationships obtained from empirical data collected at those locations (see Figure S1).”*

Figure S1 Legend: *“Note that the relationships were constructed from empirical length and weight data collected at the very same sites.”*

REVIEWERS' COMMENTS

Reviewer #1 (Remarks to the Author):

I appreciate the authors' clarification in text addressing my previous comments. Their revisions improve the context of their findings and process, in my opinion. As the authors have adequately addressed my previous suggestions, I have nothing further to suggest.

Reviewer #2 (Remarks to the Author):

The authors have made significant improvements to the manuscript.

They have replaced the word "experiment" with "assay," which serves to clarify that the "chronic exposure" treatment was an in situ measurement of metabolic rate rather than an experimental manipulation.

They completed an additional analysis of effects of source temperature on metabolic rate, as requested, and include results in the supplementary information.

They changed figures to have fixed axis ranges which allows the reader to compare data values and ranges across panels and facets.

They made significant efforts to clarify the hypotheses tested and which were supported by the data, particularly on Lines 129-135 in the Tracked Changes version of the manuscript.

There the authors no longer claim support for their second hypothesis "that chronic temperature exposure reduces the temperature sensitivity of metabolic rate." This interpretation is in keeping with the result that chronic and acute exposure treatments had overlapping confidence intervals. But this result may call to question the take-home message / title as it currently reads - "Chronic exposure to environmental temperature negates the thermal sensitivity of salmonids." This title, to me, inherently implies having found a statistically significant difference between those fish chronically exposed and those not chronically exposed. This study did not find support for that pattern.

It is now clear that the main result of the study is that the chronic exposure treatment, i.e. fish measured in situ in their home environment, lacked evidence for a temperature dependency of metabolism. This partial failure to reject null hypothesis 1 (Line 108 in Tracked Changes version) - that metabolic rates do not increase with temperature - is similar to some other studies finding a lack of statistical evidence for nonzero temperature dependency of metabolism.

I recommend that the authors search the literature for other examples. Search terms like "metabolic compensation" may be helpful. One example is here:

Scheffler, M. L., Barreto, F. S., & Mueller, C. A. (2019). Rapid metabolic compensation in response to temperature change in the intertidal copepod, *Tigriopus californicus*. *Comparative Biochemistry and Physiology Part A: Molecular & Integrative Physiology*, 230, 131-137.

The authors should consider citing some of those studies and accordingly amending the Discussion, e.g. Line 196-199 in the Tracked Changes version, that reads "We are unaware of any study showing that assay conditions can lead to a non-significant relationship between temperature and metabolic rate despite the multitude of studies analysing the effects of temperature."

REVIEWER COMMENTS

Reviewer #1 (Remarks to the Author):

I appreciate the authors' clarification in text addressing my previous comments. Their revisions improve the context of their findings and process, in my opinion. As the authors have adequately addressed my previous suggestions, I have nothing further to suggest.

Response #1: Thank you for your comment. As there are no further suggestions from you, no changes have been made to the document in response to reviewer1's comments.

Reviewer #2 (Remarks to the Author):

The authors have made significant improvements to the manuscript.

They have replaced the word "experiment" with "assay," which serves to clarify that the "chronic exposure" treatment was an in situ measurement of metabolic rate rather than an experimental manipulation.

They completed an additional analysis of effects of source temperature on metabolic rate, as requested, and include results in the supplementary information.

They changed figures to have fixed axis ranges which allows the reader to compare data values and ranges across panels and facets.

Response #2: Thank you for the assessment of our manuscript and the helpful comments which we respond to below.

They made significant efforts to clarify the hypotheses tested and which were supported by the data, particularly on Lines 129-135 in the Tracked Changes version of the manuscript. There the authors no longer claim support for their second hypothesis "that chronic temperature exposure reduces the temperature sensitivity of metabolic rate." This interpretation is in keeping with the result that chronic and acute exposure treatments had overlapping confidence intervals. But this result may call to question the take-home message / title as it currently reads - "Chronic exposure to environmental temperature negates the thermal sensitivity of salmonids." This title, to me, inherently implies having found a statistically significant difference between those fish chronically exposed and those not chronically exposed. This study did not find support for that pattern.

Response #3: According the suggestion of the reviewer and editor, the title of the manuscript has been modified. We have replaced the word negation by attenuation, which is more appropriate for our study. Thus, the title of the corrected version of the manuscript is: "Chronic exposure to environmental temperature attenuates the thermal sensitivity of salmonids".

The word *negate* has also been changed by *attenuate* when it had been used in the text for this purpose.

It is now clear that the main result of the study is that the chronic exposure treatment, i.e. fish measured in situ in their home environment, lacked evidence for a temperature dependency of metabolism. This partial failure to reject null hypothesis 1 (Line 108 in Tracked Changes version) - that metabolic rates do not increase with temperature - is

similar to some other studies finding a lack of statistical evidence for nonzero temperature dependency of metabolism.

I recommend that the authors search the literature for other examples. Search terms like "metabolic compensation" may be helpful. One example is here:

Scheffler, M. L., Barreto, F. S., & Mueller, C. A. (2019). Rapid metabolic compensation in response to temperature change in the intertidal copepod, *Tigriopus californicus*. *Comparative Biochemistry and Physiology Part A: Molecular & Integrative Physiology*, 230, 131-137.

The authors should consider citing some of those studies and accordingly amending the Discussion, e.g. Line 196-199 in the Tracked Changes version, that reads "We are unaware of any study showing that assay conditions can lead to a non-significant relationship between temperature and metabolic rate despite the multitude of studies analysing the effects of temperature."

Response #4: We have considered the study provided by the reviewer and accordingly we have removed from the review the sentence relating to "*We are unaware of any study showing that assay conditions can lead to a non-significant relationship between temperature and metabolic rate despite the multitude of studies analyzing the effects of temperature.*" Instead we have added the following paragraph, which is more appropriate:

Ln192: "*The comparative lowering of respiration rates in warmer environments is referred to as metabolic compensation and temperature-independent on metabolism has been demonstrated for different aquatic organisms such as fish³¹ or copepods³², among others. At the same time, numerous previous studies have shown the dependence of temperature on metabolism on several aquatic organism^{10,33} and therefore it is essential to carry out research such as our study to analyse in depth the thermal sensibility of organism in warmer environments*".